# Train Braking Time Variations Changing the Pressurized Air Temperature

José González *[ID] and Andrés Suárez

Área de Mecánica de Fluidos (Fluid Mechanics Group), Universidad de Oviedo, C/Wifredo Ricart s/n, 33204 Gijón, Spain; brigadier.asf@gmail.com
* Correspondence: aviados@uniovi.es; Tel.: +34-985-18-20-99

**Abstract:** Braking time in a moving train at standard speeds has become a critical variable that increasingly concerns the industry. The present paper discusses the possible option of temperature variation to cut down the response time of the whole pneumatic braking system in a train installation. A pneumatic system, considered equivalent to the system existing in a real train, was experimentally analyzed to account for the time and characteristics of a sonic pressure wave moving in the pipes. The available system behavior was compared for two different air temperatures. The obtained results point to a relevant temperature effect on the pressure wave transmission, which may promote time or distance shortening in a standard braking process. Although in the experimental campaign only two initial temperatures could be set, the study shows a possible research path for future improvements. A parallel theoretical calculation corrected by the effect of the relevant elements in the pipes was performed to allow a comparison with the experiments.

**Keywords:** trains; braking system; energy transmission; pneumatic pipeline; pressure wave; detention time; air-based systems





## 1. Introduction

Modern trains are among the safest modes of transportation currently in use. Nevertheless, referring to their main braking systems, many improvements are still possible. The quest for the optimal braking system can be traced back to the middle of the 19th Century, when demands for speed in trains were growing and the industry was still developing early solutions (a quite interesting overview of the old and up-to-date braking methods is depicted in [1] and updated in [2]). In fact, the first solutions involved the direct effect of the direct human force to the more advanced method, which applied the use of the pneumatic force as a first improvement; see [3,4].

The first pneumatic solution in the railway industry is dated to around the year 1869, when Westinghouse patented the first and very efficient pneumatic brake (see, for instance, https://es.wikipedia.org/wiki/George_Westinghouse, last accessed 24 May 2021, or [5]). The basic principle is that compressed air pushes on a piston in a cylinder. The piston is connected to a brake shoe, which can rub on the train wheel and by creating friction, finally stopping the train. Therefore, from the early introduction of the pneumatic solutions, the compressed air was conducted throughout a piston-driven brake acting on the wheels of the car in the train (see sketches in [1]).

On the other hand, the development of mechanical and electrical systems has encouraged designers, increasing the system's performance over time. Combining the evolution of the mechanical, electrical, and pneumatic systems, together with an increase in security due to regulation limits (see Union Internationale des Chemins de Fer, UIC, Standards, for instance [6,7], among others), major improvements have been introduced in the train braking system. However, the problem of braking time and distance has still been an issue in the last several years. All management protocols are based on a set of supervision curves, which correlate to the allowed maximum speed of the train and the distance covered before

it comes to a halt. This last fact also addresses some possible actions on the emergency braking system in the case that the actual train speed is over the allowed one, as shown in [8].

From the previously mentioned studies, and probably many other important contributions ([9–11], among them), there is a clear interest in improving the knowledge about braking systems, particularly stressing the need for a time reduction in the operation of the train cars in order to promote their final stoppage. Some of the global questions are yet to be solved, such as the calculation method for the braking time or distance and the possible measures to decrease both with the technology available. Particularly [9,10] studied the stiffness variation of a bogie and the forces during the train braking period, respectively.

As stated, the reliability of the methods to evaluate the braking of freight trains is one of the most important issues enabling the interoperability of railway transport in the EU network [12]. The existing official or authority rules for braking calculations are the MPS for Traction Calculations (Russian government) and the TSI (Technical Specifications for Interoperability), which were approved by the EU Commission in 2006 (2006/861/EC), modified in 2009, and latterly amended by 2012/464/EU. An interesting article about braking delay time and signal transmission was published, including a study on the forces, Ref. [4].

Some other options available in the literature range from innovative solutions that include the gradient averaging concept [13] to even more recent and sophisticated mathematical tools and solutions (for instance, refs. [14–16]). The work using the Hopf bifurcation theory introduced by [15], which shows a stability analysis for the braking signal and its effects on the whole system, is also remarkable. Although the approach of the Hopf is quite advanced and opens new horizons, experimental measurements should always support such theories.

The different improvements in the railway industry toward an optimized braking system have been conditioned by the specific analysis of the train–railway interaction and driven by the catastrophic consequences of accidents. It is convenient that any possible improvement will based on the currently used systems, together with a technical and economical feasibility study.

In the present paper, an experimental procedure to obtain a reliable improvement in the braking time and distance is shown. The preliminary ideas and description of the present study were developed under a more global study to redefine the time–distance braking conditions for standard-speed trains [17]. This approach is based on the compressed air brake and does not include the more modern and recent high-speed trains, which normally include more sophisticated solutions, namely magnetic-based brakes [18]. Nevertheless, to the authors' knowledge, the possible option of temperature influence in the braking time has not been widely studied.

## 2. Experimental Set-Up and Measurements

The typical air supply for an air-based braking system in a train (for a wider range of options, see [19]) includes the driver's brake valve, an air supply, a common brake pipe for the different cars, the brake cylinders connected to the brake shoes, and, finally, the car wheels. The pneumatic system advantages are well-known and cover items such as the flexibility of the installations; cost reductions in the generation, transport, and storage; and the lack of fire and explosion problems.

Considering global pneumatic brake systems, the final braking and stopping time becomes a sum of the following:

- Driver's order.
- Pneumatic transmission of the order or brake delay time.
- Brake shoe action on the wheel.
- Wheel and train final stopping.

All these times can also be converted into distances, Ref. [20]. Either in time or distance before stopping, there is still a certain level of human effect on the whole process.

On the other hand, many technical improvements have already been introduced, so the only possible gain might come from the pneumatic order transmission and pneumatic installation. For instance, in the standards, the filling time established by the UIC is approximately 2.5 s, and the driving force from the driver's act on the braking valve is then transmitted along the main brake pipe. The final braking distance depends on many factors. The most important ones were already listed in many previous works [13]. From such lists, the brake time delay comes to play a relevant role, as little or no improvement can be implemented on the other factors. The brake time delay is defined as the time for the pneumatic system to act from the driver command up to the time in which effective braking is transmitted to the wheel.

From previous knowledge and keeping in mind some of the limitations already set in the bibliography, such as the pressure gradient limit [15] and speed evolution as a function of the friction coefficient, the challenge of the time reduction for the whole process is still present [17]. Considering the different variables involved in pressure wave transmission, a possible gap is found for improvement in consideration of the temperature.

With the idea of focusing on finding an experimental value of the possible change in the brake performance as a function of temperature, a series of tests were carried out in the Berrón Integral Maintenance Site (BIMS). This facility, owned by RENFE (Spanish Railway Company) and located at $43°22'57''$ N and $5°42'09''$ W, is used to check and continuously review the state of the main train parts: locomotive engine, cars, equipment, and so on. Preventive maintenance protocols are widespread for the different components. Apart from in-house reviewing of the trains, post-sale services are offered to different companies at this site.

Therefore, the idea behind the experiments is to perform them in an installation as similar as possible to a real train. To perform the different tests, a pneumatic system is available at the BIMS with a long piping system to simulate the same effect as the one existing in a real train. The facility is meant to allow a full check of the train units, and, therefore, the total length is more than 500 m from the beginning to the end of the circuit (see Figure 1).

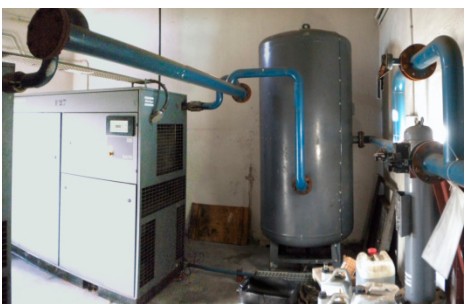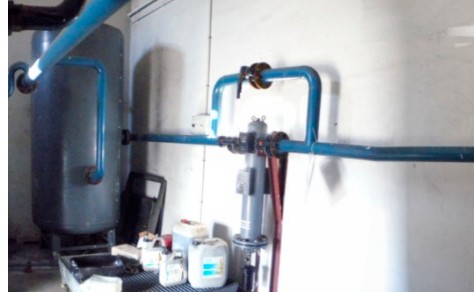

**Figure 1.** Piping system for the pneumatic transport.

The only element not considered in the experiments is the coupling between two consecutive wagons. Usually, such coupling is performed with a draw gear and a damper [21] so that the whole coupling behaves as a non-linear spring with a variable stiffness-to-damping ratio and with dead zone and end runs. In addition, regarding the pipe, this is solved with a pair of flexible coupled hoses. This effect typically becomes important when the number of wagons is high (higher than 20, for instance) because of the superposition of different pressure waves, as can be observed in [22].

A sketch of the defined system is shown in Figure 2, where the main pneumatic elements are displayed in a symbolic description. The main elements are the filter (#1), the compressor (#3), and the FCL (filter, control, and lubrication elements, with numbers #7, #8, and #10), apart from the operation valves $V_1$ and $V_2$, placed at the start and end of the test circuit. Valve #12 sets the operation of the system for a braking procedure, and it is

named $V_0$ in what follows. In addition, the pipe between valve #12 (or $V_0$) and $V_1$ will be of paramount importance in what follows, as these are the main measuring locations.

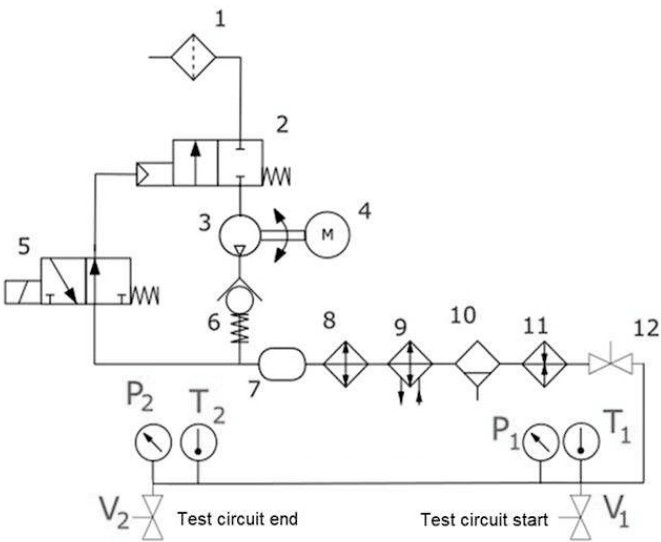

**Figure 2.** Pneumatic system schematically represented.

On the other hand, considering the block brake friction coefficient, a typical function considered by several authors is the Karwatzki friction coefficient law (see [19] or [23]), which is expressed as follows:

$$\mu_k(v, Q) = 0.6 \left( \frac{\frac{16}{g}Q + 100}{\frac{80}{g}Q + 100} \right) \left( \frac{v + 100}{5v + 100} \right) \tag{1}$$

From this definition, an evolution of the friction coefficient as a function of the speed can be obtained for a known car. The whole process allows a final running distance calculation by the integration of the ODE (ordinary differential equation) that is found. Several other processes are also available in the bibliography. In this frame, the main goal of a management railway system is based on a set of curves that relate the permitted velocity on the track and the running distance in order to ensure speed restrictions on the track line.

In the present approach, two different friction coefficients are considered (similarly to the equations shown in [16]): one for the wheel–rail contact and another for the block–wheel contact, $\mu_s$ and $\mu_k$, respectively, as sketched in Figure 3. From the two forces P and Q, together with the condition for the non-slip braking, a differential equation for the displacement or differential length is obtained as:

$$ds = \frac{-\lambda P \, v \, dv}{g \, \mu_K \, Q} \tag{2}$$

The typical assumption for the correction of the forces in the wheel–rail contact is the adhesion effect, which is caused by the elastic deformation of the wheel and rail due to the high pressure in the wheel–rail contact, giving rise to the skidding or added braking surface (considering the guidelines in the UNE Standard, 2006). The procedure to draw Figure 4 requires solving differential Equation (2) in steps for the train velocity. It becomes an iterative procedure starting at 100 km/h and reaching the behavior of the freight braking when the train halts. In the example shown, there is a blockage condition at nearly zero speed of the train. The evolution of the differential equation is represented by the arrow at the bottom of the graph. The non-slip braking condition is always assured, even in the poorer rail conditions. A complement to the present theory was firstly developed in [16], in which the bogie and body inertia were extensively considered.

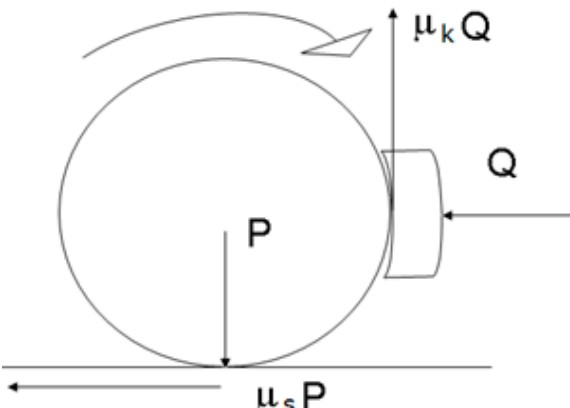

**Figure 3.** Sketch of the two main forces in a train wheel.

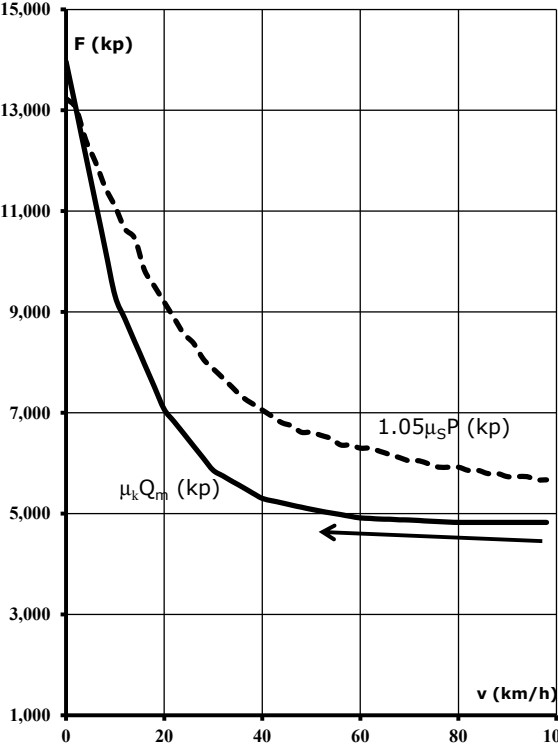

**Figure 4.** Typical force evolution (for a value of gauge pressure between 8.46 and 11.5 bar in the block–wheel contact).

From Figure 4, an iterative procedure is then established to obtain a non-sliding condition for the brake in the train as a function of the weight and horizontal force on the shoe. This procedure is highly case-dependent, and no global result can be generalized. It starts with an initial value for the weight of the different train coaches and then with a proper value of the force Q. If this value multiplied by the friction coefficient reaches the weight multiplied by the horizontal friction coefficient, namely ($\mu_k$ Q) > ($\mu_s$ P) in the calculation, this indicates wheel blockage and a dangerous slippery condition. Therefore, the blockage condition must be avoided, and the calculation is then performed using (2). In addition, to be within some security margin calculation, a 5% increase in the total load is imposed (1.05 P is the typical value used in the equation, which corresponds to the rotating mass of a standard vehicle). The known evolution for $\mu_s$ and $\mu_k$ (such as the one shown on the right-hand side of Figure 4) is then used in the procedure. The slip condition in that graph means a crossing of the two plotted curves.

The known parameters (for a given practical situation) in Equation (2) are $\lambda$, P, and g. The variables are v, $\mu_K$, and Q. As some of them are variable as a function of the train speed, integration should be performed for different time steps (corresponding to speed steps in which the parameters can be considered constant or an average value describes the system behavior), and a graph similar to the one in Figure 4 should be known from experiments or previous knowledge of the installation. With such a graph, the integration in time steps is then possible.

Although a wider and more detailed explanation of the previously mentioned procedures was published in a Ph.D. thesis work, which is the origin of the present paper, [17], a quite extensive description of the measuring devices is summarized in what follows. As sketched in Figure 2, the measurement points were placed at the beginning and at the end of the pressurized pipe (equivalent to the braking pipe in a real train). For the measurements, pressure, temperature, and time were recorded. Figure 5 shows the typical arrangement of the measuring devices at both the pipe's initial valve, $V_1$ (left side of Figure 5), and at the end of the pipe, valve $V_2$ (right side of Figure 5).

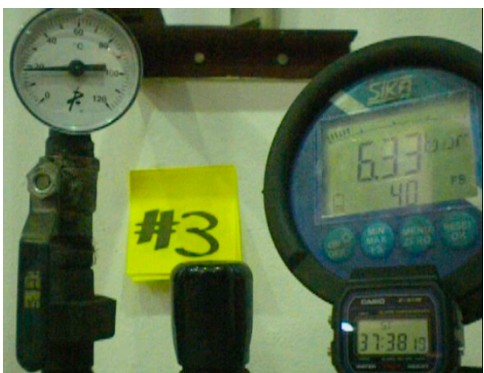 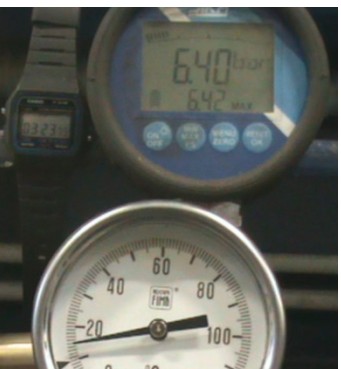

**Figure 5.** Different instrumentation ready for the tests.

Several highly sensitive instruments are required to correctly capture the compression wave that travels inside the piping system. A summary of the different features of such devices is shown in Table 1. An interesting approach was developed in [19], which shows a numerical model for the braking pipe and studies using a quasi-1D model including temperature variations. In addition, temperature dependence was attempted to be found here, and, therefore, several efforts were conducted to reach a proper value for the pressure wave transmission as a function of the temperature.

**Table 1.** Main pipe and measuring devices characteristics.

| Variable or Device | Numerical Value or Characteristics |
| --- | --- |
| Working pressure | 7.5 bar |
| Pipe diameter (steel) | 3'' |
| Immersion Thermometer | Sensitivity $\pm$ 1% for 15 bar max. |
| Contact Thermometer | Range: 0 °C to 120 °C |
| Manometers (pressure gauges) | SIKA DIGITAL<br>Max. uncertainty $\pm$ 0.005 bar. |

Different checking efforts were carried out on all the measurements, and individual repeatability tests were also carried out in order to assure the tests' validity. Nevertheless, and despite all possible uncertainty error, additional reliability of the tests comes from the fact that the comparison of any value is performed with the same set-up.

Apart from standard fluid-dynamics variables, a video recording of the different wave rebounds was operated, and the pressure wave was carefully observed, resulting in oscillations or rebounds of the pressure wave at the ends of the circuit (points 1 and 2).

Those pressure peaks were well above the uncertainty of the pressure gauges used for the measurements and, therefore, allowed an experimental value to be obtained for the sound speed in the experimental pipeline. Particularly, the pressure peaks were around 20 times the maximum uncertainty of the manometers (see Table 1).

The typical layout of the measuring devices has already been shown in Figure 5 for the initial and final pipe ends. The measuring of the different wave reflections or rebounds is critical to obtain an accurate value of the wave speed and the other variables, which are discussed in what follows.

## 3. Energy Transmission and First Discussion

The propagation of both pressure and depression waves inside a cylindrical pipe is the key issue for the understanding of the following paragraphs. The experiment's initial idea was to have access to an installation as shown in Figure 2, in which a long pipe would be monitored to examine the effect of the opening and closing of the initial and final valves ($V_1$ and $V_2$ in Figure 2). Temperature and pressure sensors were also available for the expected measurements, as has been previously explained. The total length that was initially expected was approximately 500 m, considering a typical train configuration.

The implemented procedure was as follows:

- The valve at the tank outlet, V0, was opened.
- Measurements were taken for the initial time, pressure, and temperature in position 1.
- Measurement was taken for the first pressure wave approach to position 2.
- Successive time and pressure increase in the pressure wave reflections were recorded.

The experimental procedure was conducted with two different working temperatures in the initial part of the setup. In fact, several additional elements, such as elbows and tees, existed in the real installation. Nevertheless, the existing tees along the pipe connected with the closed parts of the system, so they were not considered. Figure 6 shows the sketch of the actual installation used in the experiments. The number of elbows is important, as in the forthcoming calculations, an equivalent length is obtained.

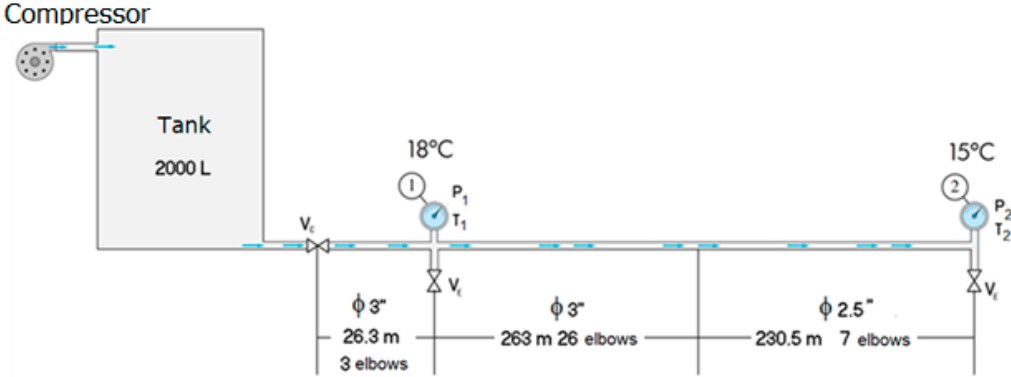

**Figure 6.** Actual installation available at the BIMS.

The final lengths are also shown in Figure 6, together with the number of elbows in the system. A slightly higher value than 500 m was available, and when adding up the equivalent length of the elbows, the pipe's total length was even higher. As will be also mentioned later, the calculation with or without the equivalent length of the elbows introduces a slight difference in the theoretical results for the wave transmission.

In conclusion, the available installation (Figure 6) reasonably resembles the real situation in a given train. Particularly, the diameters and the installation length are properly represented in the tested geometrical arrangement. Additional elements, such as dampers in the wagon-to-wagon connections, might be considered equivalent to a set of elbows and, therefore, are also included in the set-up.

## 4. Temperature Influence and Discussion

In the frame of the previously mentioned braking conditions and with the explained installation, a baseline measurement at atmospheric temperature was performed to obtain the main pressure wave conditions: the time and reflections for a sudden valve opening at the initial point of the system. The refrigerated part of the piping system is shown in Figure 7.

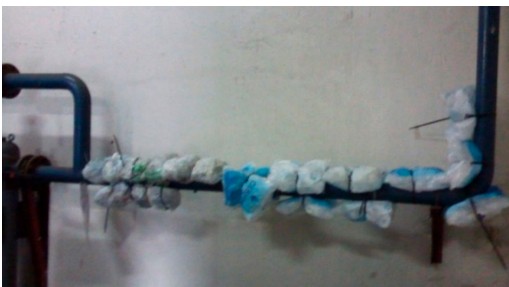

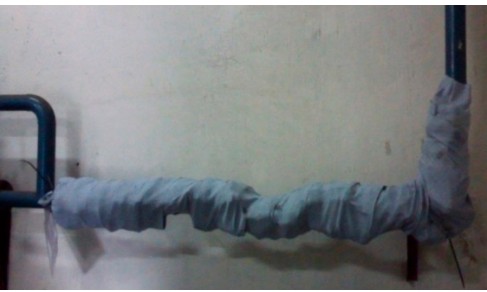

**Figure 7.** Part of the refrigerated installation with ice bags to reduce temperature, from [17], with permission.

The main equation for the pressure wave transmission or speed of sound in a pipe (as included in many heat transfer or fluid books, for instance, [24] or [25]) is:

$$c = \frac{1}{\sqrt{\rho\left(\frac{1}{K} + \frac{d}{wE}\right)}} = \frac{1}{\sqrt{\frac{p}{RT}\left(\frac{1}{K} + \frac{d}{wE}\right)}} \tag{3}$$

Therefore, for the existing pipes and performing the calculation, two different values are obtained according to:

$$\begin{cases} c\,(\text{at } 18\ ^\circ\text{C}) = 123.80\,\text{m/s} \\ c\,(\text{at } 14\ ^\circ\text{C}) = 122.94\,\text{m/s} \end{cases} \tag{4}$$

These values are obtained for a typical air Elastic Modulus (K), the Young's Modulus for the steel (E, no temperature effect considered for this value), and a width of 7 mm for the pipes. The final result gives a difference of up to 0.86 m/s or 0.7%.

Those are theoretical results, which consider neither the shape of the elbows in the flow turning (Figure 6 or Figure 8) nor the pressure wave reflections due to the end/beginning of circuit boundary conditions. The latter is a direct consequence of the pipe elasticity and fluid compressibility effects, which affect the piezometric head of the flow inside the pipeline (brake pipe). According to Figure 8, a temperature reduction of 4 °C was obtained by cooling down the air in the pipe before the inlet or first measuring position. The difference between Figures 6 and 8 is the cooling system included between valve $V_0$ and valve $V_1$ (or initial point, 1). This temperature difference is the one that imposes the two working conditions. Likewise, it is important to note that the final boundary condition on the temperature, namely on point 2, was the same in both tests.

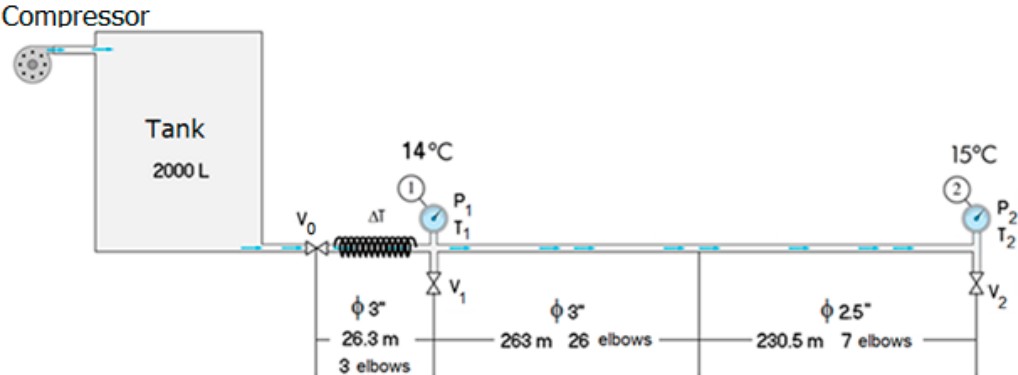

**Figure 8.** Sketch of the installation with the refrigeration system.

Considering that the pipe inlet (point 1 in Figures 6 and 8) was at a gauge pressure of 6.73 bar, the values for the rigid plastic pipe elastic modulus (E), and the bulk modulus of the air (K), the resulting values for the speed of sound for the two considered temperatures were found to be within the expected or theoretical range. Therefore, the setup was the same for both sets of measurements (temperatures), with the only difference being the installation of the cooling system and the subsequent effect of decreasing the temperature to 14 °C.

A set of experiments was conducted recording the relevant variables and keeping in mind the procedures given in the Spanish Standards, Ref. [26]. Care was taken to plot the variables in both graphs (at the right and the left) using the same scale to allow for quick comparison. The graphs show different variables, particularly fluid variables (density and two pressures, at the inlet and the outlet). The main graph plotted is the time for the wave to reach the measurement positions. This graph is plotted in a dashed line in the figures and is to be read at the left of each graph, y-coordinate. The other variables, pressure and density, are plotted in full lines and with the secondary y-coordinate (to be read at the right of each graph).

Typically, as can be observed in Figure 9, the first 5–8 experiments showed a kind of erratic or fluctuating values. Apparently, they could be assimilated to the startup of the whole system, but in fact, the whole system was already put into the nominal conditions, and, therefore, the whole set of values was considered to be valid. Furthermore, the repeatability of the signal was also considered. Therefore, no such transient behavior was visible until the whole set of elements started, and the whole set of values corresponded to working at their nominal conditions for the given temperature. From that point on, the remaining tests showed a coherent variable evolution that may be used to validate the overall calculations. At the left-hand side of Figure 8, the results are shown for the ambient temperature of 18 °C, and at the right, the results are shown for the cooled tests at 14 °C. Slight variations in the values of the pressure in 1 and 2 can be observed. In addition, an almost constant density value can be compared in the two graphs. The recorded time for the wave to move from 1 to 2 was also in the same range for the two tested temperatures.

From the results in Figure 9, it can be observed and anticipated that the time for the different measurements at 14 °C is higher than the results at 18 °C. Keeping that in mind, a statistical calculation was performed, calculating the average and standard deviation of both sets of measurements.

As stated, and using the measured values in Figure 9, the following statistical analysis can be performed, and the main results were found to be on the total time for the opening of the valve to be felt at the end of the pipe:

$$\text{At } 14\,^\circ\text{C}\begin{cases} \overline{\Delta t} = \sum_{i=1}^{N} \Delta t_i = 14.20\,\text{s} \\ \sigma = \sqrt{\frac{1}{N-1}\sum_{i=1}^{N}\left(\Delta t_i - \overline{\Delta t}\right)^2} = 7.68\,\text{s} \end{cases}$$

(5)

$$\text{At } 18\,^\circ\text{C}\begin{cases} \overline{\Delta t} = \sum_{i=1}^{N} \Delta t_i = 13.71\,\text{s} \\ \sigma = \sqrt{\frac{1}{N-1}\sum_{i=1}^{N}\left(\Delta t_i - \overline{\Delta t}\right)^2} = 7.71\,\text{s} \end{cases}$$

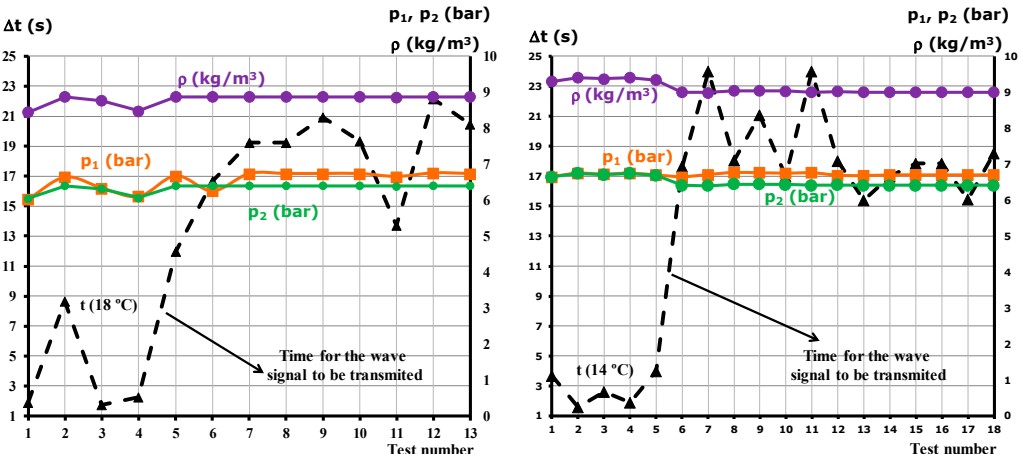

**Figure 9.** Results for different tests: time, pressures, and densities (left 18 °C and right 14 °C). Modified from original figures in [17], with permission.

From the experiments, two main fluid variables were obtained, namely the density and the speed of sound. These two variables, together with the time for the first peak to reach the measuring locations, are plotted in Figure 10. Again, as in Figure 9, at the left-hand side, the resulting tests for the inlet temperature of 18 °C are plotted, and on the right-hand side, those for 14 °C are plotted. In addition, the same scale was kept between the left- and right-hand sides of the graph to allow for comparison.

In the comparison of both graphs in Figure 10, a slight decrease in the sound speed is found when changing the temperature. In particular, a decrease in its value is clearly observed. Considering the video recording of the different wave reflections and bounces, a typical value of 6 to 10 bounces can be clearly identified. The whole time for the different pressure waves generated when opening the valve in point 1 was considered to obtain the average speed of sound value.

According to Figure 10, it can be observed that the tests conducted at the cooled temperature for the inlet flow promoted a lower speed of sound; the average of the plotted values was 124.86 m/s. This value was compared with the one for the baseline pipe, which had an average value of 127.06 m/s. The difference obtained was 1.73% and can be considered too low, but it is still relevant for this kind of test. Additionally, although it may be considered too low, as will be shown in the next paragraph, the gain in braking length was much higher.

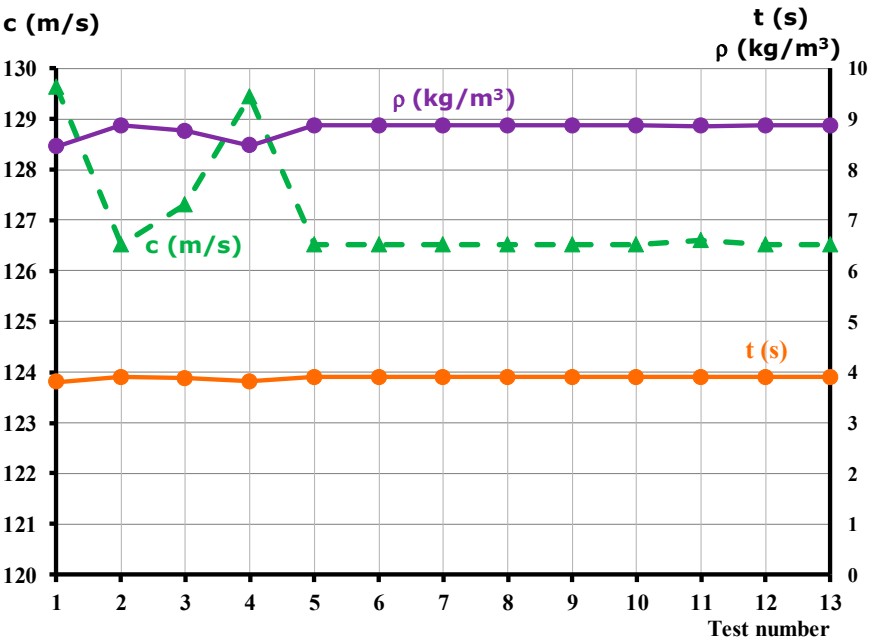

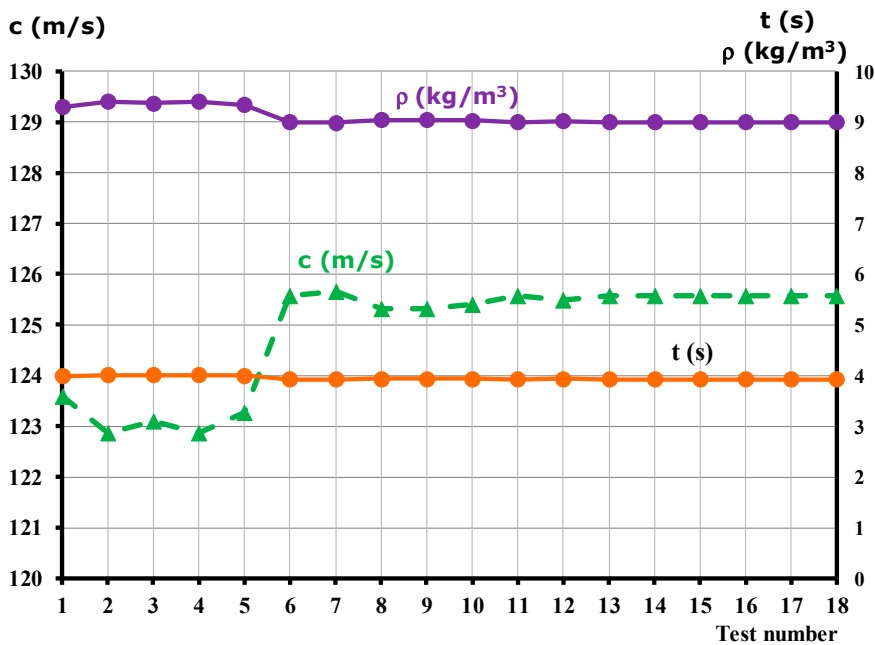

**Figure 10.** Results for different tests: time, different tests: time, density, and speed of sound (upper Figure: 18 °C and lower Figure: 14 °C).

A statistical calculation on the values of Figure 10 gives the following results:

$$
\text{At } 14\,^{\circ}\text{C} \begin{cases} \bar{t} = \sum\limits_{i=1}^{18} t_i = 3.953\,\text{s} \ , & \bar{c} = 124.859\,\text{m/s}, \ \bar{\rho} = 9.110\,\text{kg/m}^3 \\[2mm] \sigma_t = \sqrt{\frac{1}{17}\sum\limits_{i=1}^{18}(t_i - \bar{t})^2} = 0.0348\,\text{s}, \ \sigma_c = 1.109\,\text{m/s}, \ \sigma_\rho = 0.164\,\text{kg/m}^3 \end{cases}
$$

$$
\text{At } 18\,^{\circ}\text{C} \begin{cases} \bar{t} = \sum\limits_{i=1}^{13} t_i = 3.884\,\text{s} \ , & \bar{c} = 127.057\,\text{m/s}, \ \bar{\rho} = 8.798\,\text{kg/m}^3 \\[2mm] \sigma_t = \sqrt{\frac{1}{12}\sum\limits_{i=1}^{13}(t_i - \bar{t})^2} = 0.0354\,\text{s}, \ \sigma_c = 1.120\,\text{m/s}, \ \sigma_\rho = 0.152\,\text{kg/m}^3 \end{cases}
$$

(6)

The standard deviations for each variable were almost the same for the two temperatures. This could indicate a good experiment design.

Correspondingly, the time or distance was obtained using the values experimentally found and applying the procedure explained in Equation (2). A comparison with the theoretical approach is shown in Figure 11. Three different values for the two temperatures were plotted: theoretical, theoretical corrected, and experimental.

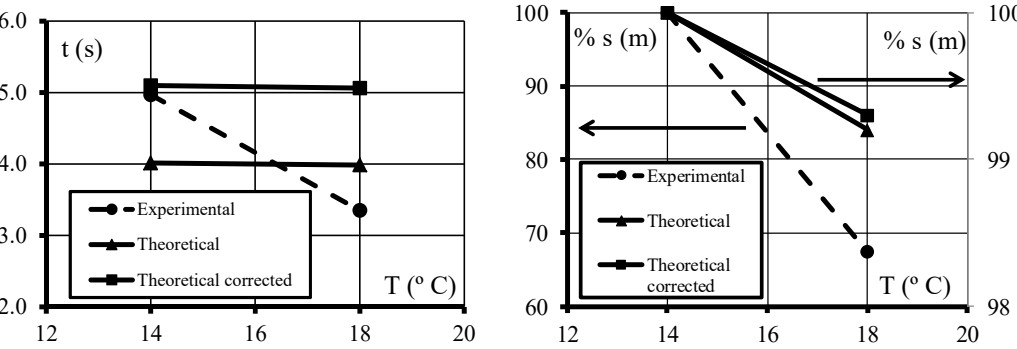

**Figure 11.** Evolution of breaking time and time and length as measured in the two temperature conditions and comparison with theoretical values.

The theoretical value was found using the values from (4) and considering the lengths of the piping system in the installation and measuring only the first wave reaching from 1 to 2. They are plotted for the two temperatures and follow a slightly decreasing trend with the temperature.

The values in Figure 11 for the theoretical corrected values were different from the theoretical values because for the theoretical value, there was no correction for the equivalent length of the elbows, and only the first pressure wave arriving was considered to obtain the speed of sound. As can be observed in Figure 11, they differed slightly from the theoretical ones, with a similar trend. An initial speed for the studied train equal to 54 km/h was considered for the theoretical and theoretical corrected calculations.

Finally, the experimental values plotted in Figure 11 (left-hand side) were obtained from the previous graphs (Figures 8 and 9), from which the sound speed was directly measured. The average time was obtained considering the different wave bounces. Such bounces are not easily explained from Figure 8 or Figure 9 but were video recorded and post-analyzed with intensive care. An average of the times for the two temperatures provided the values shown in the graph, which correspond to the left-hand graph in Figure 11. The numerical results were obtained as follows (i stands for the test number and j stands for each rebound):

$$\begin{cases} t_{18°}^{Exp} = \dfrac{\sum\limits_{i=1}^{13} \left( \dfrac{\Delta t_j}{n_i} \right)}{13} = 3.349\,s \\[4mm] t_{14°}^{Exp} = \dfrac{\sum\limits_{i=1}^{18} \left( \dfrac{\Delta t_j}{n_i} \right)}{18} = 4.976\,s \end{cases} \tag{7}$$

One of the measurements promotes a positive temperature gradient along the whole pipe length, whereas the second one imposes a negative temperature gradient. This is another key issue for the differences in the main values shown in Figures 9 and 10. Typically, the velocity gradient is considered a variable in the propagation of a wave in a given pipe. The percentages of the braking length are then obtained with the protocol to calculate the distance according to Equation (2). As can be observed, a slight decrease was found for the theoretical values, approximately 1% of the total braking distance (see the right-hand side coordinate for the right graph in Figure 11), while a much higher value, approximately 32.6%, was obtained for the experimental values (see the left-hand side coordinate in the right graph, Figure 11). In all cases, this gain was obtained by increasing the temperature.

To summarize, the experimentally measured wave speed and the brake system answer become better at the initial temperature of 18 °C. This is in agreement with the theoretical expected values and leads to the conclusion that during winter operations, heating the system would have a positive effect.

Even though the geometrical arrangement in a train would be slightly different, the lengths and piping system were chosen in an available installation to be as close as possible to the actual braking procedure.

## 5. Conclusions

The time for the pneumatic transmission of a given braking order in a train, also called delay time, could be modified by acting on the fluid characteristics, particularly by changing the fluid density. Inherent problems in the signal treatment and experimental limitations could counterbalance a possible and feasible (at least on a theoretical basis) improvement or pressure wave acceleration. The dynamic effects of the flow in a train pneumatic system were experimentally analyzed using an equivalent installation to the one in a real train. The compressibility effects in the pressure wave transmission were considered.

During the measurement campaigns, different wave rebounds were observed, and an average wave or speed of sound was obtained for the two different temperatures. The main conclusions of the experiments point to a decrease in the braking time or length with the temperature increase. If the trend is consolidated with more data for different temperatures, a stronger and more practical conclusion can be drawn.

Globally speaking, there is a quite important agreement in the results comparing the theoretical prediction and the experiments. The main differences may come from the effect of the elbows and the inherent generation of reflections in the wave that may affect the observed rebounds. The concept of the equivalent length works very accurately for the losses, but not in the secondary effects that may cause very complex flow features, such as the ones analyzed in the present paper.

The observed results show a clear temperature dependence on the final train braking time and open a possible area for improvement by changing the value of this variable along the pressurized line, especially in winter operations, which seem to be the more critical ones. In addition, and according to the experimental results, a temperature increase would cause a shortening in the total braking time. Although the whole braking process in a train is subjected to many restrictions involving the one-way traffic signs, drivers response time, and mechanical constraints, which cannot be covered here, a possible optimization by acting on the temperature was shown.

Inasmuch as only two temperatures were analyzed, with the available data, it is not possible to assess the value of the optimum temperature. Nevertheless, with the performed measurements, a 1% decrease in the speed of sound was found, and, with the whole braking procedure, shortening lengths of up to 32.6% were found with the 4 °C change imposed. The fact that the best temperature to decrease the braking distance is the higher one indicates a better performance of the braking system in the summer than in the winter.

**Author Contributions:** Conceptualization, J.G. and A.S.; methodology, J.G.; software, J.G.; validation, A.S., formal analysis, A.S.; investigation, J.G. and A.S.; resources, J.G.; data curation, A.S.; writing—original draft preparation, J.G.; writing—review and editing, J.G. and A.S.; supervision, J.G. All authors have read and agreed to the published version of the manuscript.

**Funding:** This research received no external funding.

**Informed Consent Statement:** Not applicable.

**Data Availability Statement:** Not applicable.

**Acknowledgments:** The authors gratefully acknowledge the financial support from the "Ministerio de Ciencia, Innovación y Universidades" (Spain) under project ENE2017-89965-P and from another project sponsored by the Ministerio de Economía y Competitividad, Gobierno de España: "Tec-

nologías ecológicas para el transporte Urbano, ecoTRANS" (CDTI). In addition, the financial help from the IUTA, through projects SV-17-GIJON-1-07, SV-18-GIJON-1-04, and SV-19-GIJON-1-15, is gratefully acknowledged here.

**Conflicts of Interest:** The authors declare no conflict of interest.

### List of Symbols

| | |
|---|---|
| c | Speed of sound in the pressure wave transmission, (m/s). |
| d | Brake pipe diameter, (m). |
| E | Young's modulus of the pipe, (Pa). |
| g | Gravity acceleration, $(m/s^2)$. |
| i | Subindex to account for the different tests. |
| K | Bulk or elastic modulus of the air, (Pa). |
| n | Number of rebounds of the wave in each test. |
| p | Pressure, (bar). |
| P | Force in the contact wheel and railway, (N). |
| Q | Force in the contact block and wheel, (N). |
| R | Gas constant, (J/mol K). |
| s | Length or moving distance when braking, (m). |
| t | Time, (s). |
| T | Temperature (K) or (°C). |
| V | Valve in the pneumatic system. |
| v | Speed of the train, in (km/h). |
| w | Pipe width, (m) |
| $\rho$ | Air density, $(kg/m^3)$. |
| $\lambda$ | Inertia coefficient, (- - -). |
| $\mu_K, \mu_S$ | Friction coefficient in the wheel–railway or shoe–wheel contact. |
| $\sigma$ | Standard deviation. |
| $\Delta$ | Increment of a given variable. |

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
