# Peer review of "Train Braking Time Variations Changing the Pressurized Air Temperature"

_fluids, doi:10.3390/fluids6100351_

Round 1

Reviewer 1 Report

  1. The abstract is too cumbersome to summarize the main research contents, research methods, results and conclusions of the article

  1. How is the time component of Fig. 9 and FIG. 10 obtained? How is it obtained through pressure observation? Can you explain in detail what method is used to observe it?

  1. According to the video experimental data, how to count the bounce times of pressure wave? What method is used for statistics? (in line 350 351)?

  1. How can the time and distance of train braking be obtained through experimental data and formula (2)? Can you elaborate on the calculation process? (in line 366 367)

  1. Which parameters of formula (2) are known and which are obtained from experimental data? Those are the quantities to be asked? (in line 174 175)

Author Response

A detailed answer has been uploaded in a file

Reviewer 2 Report

The manuscript is well written and the explanation of the problems and the development of the study and the experimental part are very clear.
The topic is very current and can have great practical application.
The experimentation is well described and the results are comforting.
I suggest adding a statistical discussion of the results, which will most likely show a high repeatability of the result. The values have been shown in the graphs, but it would also be advisable to indicate the fluctuations of the results with an average value and the standard deviation, to understand how much variation is present between the recorded values.
For everything else the manuscript is well conducted and does not require further modifications or implementations.

Author Response

Find the answers in an attached file

This manuscript is a resubmission of an earlier submission. The following is a list of the peer review reports and author responses from that submission.

Round 1

Reviewer 1 Report

First I see that the authors are honest with the data.

But something is wrong: figure 9 show hypersonic signal transmission at test 1,3,4 (18ºC) and 2,4 (14ºC). My first interpretation is human error. I do not know if is bad experiment design or bad data acquisition. In my opinion your data acquisition should be done using a Digital Acquisition System (DAQ). DAQ systems allow to see the wave form and improve time precision. And today there are good low cost DAQs.

What is shown in figure 11? An incredible improvement from 14ºC to 18ºC. If your data is good, there is something more improving brake efficiency.

.....

Line 11 replace "energetic wave" with "pressure wave"

Line 12-13 "energy exchange between two states" sound strange. Better something like "transformation from kinetic energy to heat energy"

Equation 2 "P/g" is strange. Using International System of Units is preferred "m" mass. What means "rho"? I think is not air density as stated at line 456.

Equation 4,5 on my PC there is a strange symbol.

Reviewer 2 Report

1. Format of the paper shall be improved. E.g. Table 1 is splitted in two pages.
2. Formula 4 not understandable. Is it in Chinese language?
3. How the writers consider the effect of current turbulences in the braking performance?
4. Braking performance is a multidimensional problem and different parameters have mutual effects on the performance. So all these have to be considered.

Reviewer 3 Report

1.The title of the article suggests that train braking times are temperature dependent. Train braking time should be independent of temperature.

2. In my opinion, the scope of the analysis of the influence of temperature on the speed of sound within a few degrees is not representative. Even in Europe, trains are successfully operated at temperature differences above 50K. As is known, for a particular pipeline, the speed of sound is influenced primarily by air density and pressure.

3. What is the total error value of the speed of sound measurement in this stand?

Reviewer 4 Report

1. Part of the text in line 170 appears in the form of pictures, which does not meet the requirements.

2. Due to the fluidity of the liquid in the pipeline, is the cooling method of the ice bag in Figure 7 reliable? How to verify whether the stable temperature suitable for the experimental conditions has been reached?

3. The punctuation in line 87 is redundant.

4. The number of temperature samples is too small to support the experimental data.

5. Some statements are not smooth, such as lines 9 to 10 “the time or distance until complete halt is called the stopping distance......”

6. Fig. 6 Fig. 8 is not completely displayed, so it is impossible to view the installation schematic and parameters completely.